# The First Synthesis of Periodic and Alternating Glycopolymers by RAFT Polymerization: A Novel Synthetic Pathway for Glycosaminoglycan Mimics

**DOI:** 10.3390/polym11010070

**Published:** 2019-01-05

**Authors:** Masahiko Minoda, Tomomi Otsubo, Yohei Yamamoto, Jianxin Zhao, Yoshitomo Honda, Tomonari Tanaka, Jin Motoyanagi

**Affiliations:** 1Faculty of Molecular Chemistry and Engineering, Graduate School of Science and Technology, Kyoto Institute of Technology, Matsugasaki, Sakyo-ku, Kyoto 606-8585, Japan; m7618002@edu.kit.ac.jp (T.O.); m5618030@edu.kit.ac.jp (Y.Y.); jinmoto@kit.ac.jp (J.M.); 2Department of Orthodontics, Osaka Dental University, 8-1, Kuzuhahanazonocho, Hirakata, Osaka 573-1121, Japan; jianxinzhao@hotmail.com; 3Institute of Dental Research, Osaka Dental University, 8-1, Kuzuhahanazonocho, Hirakata, Osaka 573-1121, Japan; honda-y@cc.osaka-dent.ac.jp; 4Department of Biobased Materials Science, Graduate School of Science and Technology, Kyoto Institute of Technology, Matsugasaki, Sakyo-ku, Kyoto 606-8585, Japan; t-tanaka@kit.ac.jp

**Keywords:** periodic glycopolymer, alternating glycopolymer, RAFT copolymerization, carbohydrate-substituted vinyl ether, carbohydrate-substituted maleimide, glycosaminoglycan mimics, CuAAC click reaction

## Abstract

This study concerned the controlled synthesis of periodic glycopolymers by reversible addition-fragmentation chain transfer (RAFT) copolymerization. To this end, maltose- and lactose-substituted vinyl ethers (MalVE and LacVE, respectively) and maltose-substituted maleimide (MalMI) were newly synthesized. RAFT copolymerization of MalVE and ethyl maleimide (EtMI) (monomer feed ratio: MalVE:EtMI = 1:1) afforded periodic glycopolymers (poly(MalVE-*co*-EtMI)) consisting of major parts of alternating structure (-(MalVE-EtMI)_n_-) and a small part of consecutive sequences of EtMI (–EtMI-EtMI-). Occurrence of the latter sequences was caused by the homopolymerizability of maleimide under the present polymerization condition, and the formation of the consecutive sequences of EtMI was successfully suppressed by varying the monomer feed ratio. RAFT copolymerization of LacVE and EtMI was also found to proceed and similarly yielded periodic glycopolymers (poly(LacVE-*co*-EtMI)). Moreover, RAFT copolymerization of LacVE and MalMI (monomer feed ratio: LacVE:MalMI = 1:1) was performed to give copolymers (poly(LacVE-*co*-MalMI)) having composition ratio of LacVE/MalMI ≈ 36/64. The resultant periodic glycopolymers poly(MalVE-*co*-EtMI) and poly(LacVE-*co*-EtMI) were subjected to lectin binding assay using concanavalin A and peanut agglutinin, exhibiting the glycocluster effect. Moreover, these glycopolymers obtained from the copolymerization of VE and MI were found to be non-cytotoxic.

## 1. Introduction

Glycosaminoglycans (GAGs) are known as physiologically active polysaccharides being ubiquitous both in the extracellular matrix and on cell surfaces, and play key roles in wide-ranging life phenomena as well as in diseases affecting mammalians [1,2,3]. From the view of structural aspect, GAGs are long, linear, unbranched, and negatively-charged polysaccharides composed of repeating disaccharide units consisting of an amino sugar and an uronic acid. In addition, major GAGs are sulfated and distinguished from each other by the disaccharides structures and various sulfation patterns [1,2,3]. Thus, GAGs are naturally occurring alternating copolymers. The synthesis of GAG mimics based on glycopolymers by both conventional and controlled polymerizations have been reported [4,5,6,7,8,9,10]. Particularly, in the last decade, with the remarkable development in precisely controlled radical polymerization such as ATRP and RAFT polymerization techniques [11,12,13,14,15,16,17], various multi-component glycopolymers with two or more carbohydrate-substituted repeating units have been synthesized and investigated as GAG mimics [5,6,7,8]. In the syntheses of GAG-mimetic glycopolymers thus far reported, the most important subjects are the incorporation of sulfated carbohydrate moieties into glycopolymer structure and regulating of the density of the sulfate groups for exhibiting physiological functions. Consequently, sequence control of the repeating units in the multi-component glycopolymers are still beyond our reach. This fact prompted us to develop a novel synthetic strategy for forming sequence-controlled glycopolymers, more specifically, glycopolymers having alternating structure. Until now, few examples have been reported about the sequence-control in glycopolymers; however, they are not alternating structure but short-block ones [18]. To achieve the alternating structure, we focused on the copolymerization of an electron-rich and an electron-deficient vinyl monomers, which have been widely recognized to afford alternating copolymers via radical polymerization mechanism [19,20,21,22,23,24,25,26]. To this end, vinyl ethers (VEs) and maleimide (MI) monomers having pendant unprotected carbohydrate moieties were newly designed by utilizing copper(I)-catalyzed azide-alkyne cycloaddition (CuAAC) click reaction between alkyne-substituted VE and MI and disaccharide azides, respectively. Here, the employed maltosyl and lactosyl azide were obtained by Shoda’s activation of unprotected sugars employing 2-chloro-1,3-dimethylimidazolinium chloride [27,28]. Disaccharide-substituted VEs and MI correspond to electron-rich and electron-deficient vinyl monomers, respectively. In addition to controlling the repeat-unit sequences, we aimed to control molecular weight and molecular weight distribution by applying RAFT copolymerization to the synthetic protocol of glycopolymers. In this study, RAFT copolymerization of maltose-substituted VE (MalVE) and ethyl maleimide (EtMI) was firstly investigated to obtain periodic glycopolymers, where MalVE unit are distributed as every other unit, and the polymerization conditions was optimized to attain a more accurate control in alternating structure. Synthesis of periodic polymers by RAFT copolymerization of naturally occurring vinyl monomers and MI derivatives were investigated by Sato and Kamigaito to yield various periodically functionalized polymers [29,30]; however, synthesis of periodically carbohydrate-functionalized polymers (periodic or alternating glycopolymers) have not yet been reported. Therefore, we demonstrated RAFT copolymerization of lactose-substituted VE (LacVE) and maltose-substituted MI (MalMI) to obtain glycopolymers periodically carrying two kinds of disaccharide residues in the pendants. This would be the first prototype of GAG-mimetic glycopolymers focusing on the sequence-control (Figure 1). In addition, specific interactions with lectins of the resultant periodic glycopolymers were examined by lectin binding assay. To confirm the potential applicability of the glycopolymers in this study as promising materials in biomedical and pharmaceutical field, cytotoxicity assessment was performed.

## 2. Materials and Methods

### 2.1. Chemicals and Reagents

*D*(+)-maltose monohydrate (FUJIFILM Wako Pure Chemical Corporation, Osaka, Japan, 98.0%), *D*(+)-lactose monohydrate (FUJIFILM Wako Pure Chemical Corporation), sodium azide (FUJIFILM Wako Pure Chemical Corporation, 98.0%), 2-chloro-1,3-dimethylimidazolinium chloride (DMC; FUJIFILM Wako Pure Chemical Corporation, 97.0%), *N*,*N*-diisopropyl ethylamine (DIPEA; TCI, Tokyo, Japan, 98.0%), copper (II) sulfate pentahydrate (FUJIFILM Wako Pure Chemical Corporation, 99.5%), l-ascorbic acid sodium salt (AscNa; FUJIFILM Wako Pure Chemical Corporation, 98.0%), [(1-benzyl-1*H*-1,2,3-triazol-4-yl)methyl]amine (TBTA; Tokyo Chemical Industry Co., Ltd., Tokyo, Japan, 97.0%), *N*-ethylmaleimide (EtMI; Tokyo Chemical Industry Co., Ltd., Tokyo, Japan, 98%), and 2,2′-azobis-[2-(2-imidazolin-2-yl)-propane] dihydrochlroride (VA-044; FUJIFILM Wako Pure Chemical Corporation, 97%) were used as received. Pre-wetted dialysis tubing with MWCO = 1 KD (Spectra/Por 7; nominal flat width: 18 mm, diameter 11.5 mm, volume: 1.1 mL/cm, length: 10 m) (Spectrum, Rancho Dominguez, CA, USA) was performed in a 1 L beaker by changing distilled water four or five times over a period of 24 h. Fluorescein isothiocyanate (FITC)-labeled Concanavalin A (Con A) from *Canavalia ensiformis* and FITC-labeled peanut agglutinin (PNA) from *Arachis hypogaea* were purchased from J-Oil Mills. Inc. (Tokyo, Japan) All other reagents were commercially available and used without further purification. Synthesis of *N*-propargylmaleimide was carried out according to the procedure reported in the literature [31]. Maltosyl azide (MalN_3_) and lactosyl azide (LacN_3_) were prepared according to the literature [27,28]. A chain transfer agent 2-(benzylsulfanylthiocarbonylsulfanyl) ethanol (BTSE) was synthesized using 2-mercaptoethanol, carbon disulfide, and benzyl bromide according to the procedure reported in the literature [32]. Synthesis of 3-[2-(2-vinyloxyethoxy)-ethoxy]-propyne (VEEP) was carried out by a reaction of 2-(vinyloxyethoxy)-ethanol with propargyl bromide in the presence of KOH in DMSO at room temperature for 40 h [33].

### 2.2. Methods

^1^H and ^13^C NMR spectra were recorded at 25 °C on a Bruker model AC-500 spectrometer (Bruker, Billerica, MA, USA), operating at 500 and 125 MHz, respectively, where chemical shifts (*δ* in ppm) were determined with respect to non-deuterated solvent residues as internal standards. Analytical size exclusion chromatography (SEC) was performed in 0.2 mol·L^−1^ NaNO_3_ aqueous solution at 40 °C, using 7.8 mm × 300 mm gel columns (TOSOH TSKgel *α*–3000 × 3) on a JASCO model PU2089 (JASCO, Hachioji, Japan) equipped with a UV-2075 variable-wavelength UV-vis detector (JASCO) and an RI-2031 RI detector (JASCO). The number-average molecular weight (*M*_n_) and polydispersity ratio (*M*_w_/*M*_n_) were calculated from the chromatographs with respect to poly(ethylene glycol)s standards (Scientific Polymer Products, Inc., Ontario, NY, USA); *M*_n_ = 590−11,900 g/mol, *M*_w_/*M*_n_ = 1.05–1.11). Matrix-assisted laser desorption ionization time-of-flight (MALDI-TOF) mass spectrometry was performed on a BRUKER model AutoFlex III MALDI-TOF/TOF (Bruker) using 2,5-dihydroxybenzoic acid as a matrix. Fluorescence emission spectra were recorded on a JASCO Type FP-6500 spectrometer (JASCO).

### 2.3. Synthesis of MalVE

To MalN_3_ (570 mg, 1.6 mmol) was added the solution of CuSO_4_·5H_2_O (39 mg, 0.15 mmol) and AscNa (61 mg, 0.31 mmol) in H_2_O (14 mL), then the solution of TBTA (82 mg, 0.15 mmol) and VEEP (340 mg, 2.0 mmol) in DMF (14 mL) was added, and the mixture was stirred for 19 h at 25 °C. After concentration of the reaction mixture under reduced pressure, the product was purified by silica gel column chromatography (H_2_O/acetonitrile = 1/6, *v*/*v*) and then stirred with metal scavenger (SiliaMetS^®^ Imidazole, Silicycle Inc., Québec, QC, Canada)420 mg, 5 equiv. for Cu) overnight at 25 °C. After removing of metal scavenger by filtration, the filtrate was concentrated under reduced pressure, and the residue was subject to column chromatography (H_2_O/acetonitrile = 1/9, *v*/*v*) on SiO_2_, followed by freeze-dried, to allow isolation of MalVE as a white powder (530 mg, 1.0 mmol, 65%). ^1^H NMR (500 MHz, D_2_O): *δ* (ppm) 8.29 (s, 1H, triazole), 6.52 (dd, *J*_1_ = 14.3 Hz, *J*_2_ = 6.9 Hz, 1H, CH_2_=C*H*), 5.80 (d, 1H, *J* = 8.6 Hz, H1), 5.59 (d, 1H, *J* = 3.8 Hz, H1′), 4.34 (dd, *J*_1_ = 14.4 Hz, *J*_2_ = 2.2 Hz, 1H, C*H*_2_=CH), 4.16 (dd, *J*_1_ = 6.9 Hz, *J*_2_ = 2.2 Hz, 1H, C*H*_2_=CH), 4.1–3.3 (m, 20H); ^13^C NMR (125 MHz, D_2_O) *δ* (ppm) 151.9, 143.9, 123.1, 100.9, 87.2, 87.0, 79.1, 78.0, 76.6, 73.6, 73.3, 72.4, 71.6, 69.9, 69.8, 69.1, 68.9, 67.3, 63.5, 60.8, 60.3.

### 2.4. Synthesis of LacVE

To LacN_3_ (570 mg, 1.6 mmol) was added the solution of CuSO_4·_5H_2_O (39 mg, 0.15 mmol) and AscNa (61 mg, 0.31 mmol) in H_2_O (14 mL), then the solution of TBTA (82 mg, 0.15 mmol) and VEEP (340 mg, 2.0 mmol) in DMF (14 mL) was added, and the mixture was stirred for 19 h at 25 °C. After concentration of the reaction mixture under reduced pressure, the product was purified by silica gel column chromatography (H_2_O/acetonitrile = 1/6, *v*/*v*) and then stirred with metal scavenger (SiliaMetS^®^ Imidazole, 420 mg, 5 equiv. for Cu) overnight at 25 °C. After removing of metal scavenger by filtration, the filtrate was concentrated under reduced pressure, the residue was subject to column chromatography (H_2_O/acetonitrile = 1/9, *v*/*v*) on SiO_2_, followed by freeze-dried, to allow isolation of LacVE as a white powder (530 mg, 1.0 mmol, 65%).^1^H NMR (500 MHz, D_2_O): *δ* (ppm) 8.29 (s, 1H, triazole), 6.51 (dd, *J*_1_ = 14.3 Hz, *J*_2_ = 6.9 Hz, 1H, CH_2_=C*H*), 5.80 (d, 1H, *J* = 9.3 Hz, H1), 4.52 (d, 1H, *J* = 7.8 Hz, H1′), 4.35 (dd, *J*_1_ = 14.4 Hz, *J*_2_ = 2.2 Hz, 1H, C*H*_2_=CH), 4.16 (dd, *J*_1_ = 6.9 Hz, *J*_2_ = 2.2 Hz, 1H, C*H*_2_=CH), 4.1–3.3 (m, 20H); ^13^C NMR (125 MHz, D_2_O) *δ* (ppm) 151.2, 144.3, 124.3, 102.9, 88.1, 87.3, 77.7, 77.3, 75.4, 74.5, 72.5, 72.0, 71.0, 69.5, 68.9, 68.6, 67.3, 63.0, 61.1, 59.7.

### 2.5. Synthesis of MalMI

To MalN_3_ (400 mg, 1.0 mmol) was added the solution of CuSO_4·_5H_2_O (27 mg, 0.10 mmol) and AscNa (44 mg, 0.15 mmol) in H_2_O (10 mL), then the solution of TBTA (59 mg, 0.10 mmol) and *N*-propargylmaleimide (180 mg, 1.3 mmol) in DMF (10 mL) was added, and the mixture was stirred for 15 h at 25 °C. After concentration of the reaction mixture under reduced pressure, the residue was subject to column chromatography (H_2_O/acetonitrile = 1/6, *v*/*v*) on SiO_2_, (H_2_O/acetonitrile = 1/3, *v*/*v*) on Al_2_O_3_, and (H_2_O/acetonitrile = 1/9, *v*/*v*) on SiO_2_, followed by freeze-dried, to allow isolation of MalMI as a white powder (190 mg, 0.35 mmol, 35%). ^1^H NMR (500 MHz, D_2_O): *δ* (ppm) 8.23 (s, 1H, triazole), 6.90 (s, 2H, maleimide), 5.75 (d, 1H, *J* = 8.6 Hz, H1), 5.49 (d, 1H, *J* = 3.8 Hz, H1′), 4.00–3.27 (m, 14H); ^13^C NMR (125 MHz, D_2_O) *δ* (ppm) 172.2, 143.1, 134.6, 123.5, 99.6, 87.3, 77.4, 76.3, 75.7, 72.8, 72.7, 72.1, 71.7, 69.3, 60.5, 60.4, 32.1.

### 2.6. Copolymerization of MalVE and EtMI under Conventional Radical Polymerization Conditions

Conventional radical copolymerization of MalVE and EtMI was carried out with VA-044 as an initiator. To a solution of MalVE (38 mg, 70 μmol) and EtMI (8.8 mg, 70 μmol) in H_2_O (0.16 mL) and acetonitrile (0.12 mL) was added VA-044 (0.5 mg, 1.4 μmol) in a glass tube ([MalVE]_0_/[EtMI]_0_/[VA-044]_0_ = 50/50/1). The resulting solution was degassed by three freeze–pump–thaw cycles, then the glass tube was sealed under vacuum, heated at 60 °C for 1 h and quenched by rapid cooling. The reaction mixture was analyzed by SEC and ^1^H NMR spectroscopy. The products were purified by dialysis against distilled water and freeze-dried to give copolymer.

### 2.7. Copolymerization of MalVE and EtMI under RAFT Polymerization Conditions

RAFT copolymerization of MalVE and EtMI was carried out with BTSE as a chain transfer agent and VA-044 as an initiator. To a solution of MalVE (220 mg, 410 μmol), EtMI (51 mg, 410 μmol) and BTSE (2.0 mg, 8.2 μmol) in H_2_O (0.97 mL) and acetonitrile (0.72 mL) was added VA-044 (2.6 mg, 8.2 μmol) in a glass tube ([MalVE]_0_/[EtMI]_0_/[VA-044]_0_/[BTSE]_0_ = 50/50/1/1). The resulting solution was degassed by three freeze–pump–thaw cycles, then the glass tube was sealed under vacuum, heated at 60 °C for 15–120 min and quenched by rapid cooling. The reaction mixture was analyzed by SEC and ^1^H NMR spectroscopy. The products were purified by dialysis against distilled water and freeze-dried to give copolymer.

### 2.8. Copolymerization of LacVE and EtMI under RAFT Polymerization Conditions

RAFT copolymerization of LacVE and EtMI was carried out with BTSE as a chain transfer agent and VA-044 as an initiator. To a solution of LacVE (81 mg, 150 μmol), EtMI (19 mg, 150 μmol) and BTSE (0.7 mg, 3.0 μmol) in H_2_O (0.34 mL) and acetonitrile (0.26 mL) was added VA-044 (1.0 mg, 3.0 μmol) in a glass tube ([LacVE]_0_/[EtMI]_0_/[VA-044]_0_/[BTSE]_0_ = 50/50/1/1). The resulting solution was degassed by three freeze–pump–thaw cycles, then the glass tube was sealed under vacuum, heated at 60 °C for 10–80 min and quenched by rapid cooling. The reaction mixture was analyzed by SEC and ^1^H NMR spectroscopy. The products were purified by dialysis against distilled water and freeze-dried to give copolymer.

### 2.9. Copolymerization of LacVE and MalMI under RAFT Polymerization Conditions

RAFT copolymerization of LacVE and MalMI was carried out with BTSE as a chain transfer agent and VA-044 as an initiator. To a solution of LacVE (60 mg, 110 μmol), MalMI (56 mg, 110 μmol) and BTSE (0.6 mg, 2 μmol) in H_2_O (0.65 mL) and acetonitrile (0.50 mL) was added VA-044 (0.4 mg, 1 μmol) in a glass tube ([LacVE]_0_/[MalMI]_0_/[VA-044]_0_/[BTSE]_0_ = 50/50/0.5/1). The resulting solution was degassed by three freeze–pump–thaw cycles, then the glass tube was sealed under vacuum, heated at 60 °C for 20–240 min and quenched by rapid cooling. The reaction mixture was analyzed by SEC and ^1^H NMR spectroscopy. The products were purified by dialysis against distilled water and freeze-dried to give copolymer.

### 2.10. Lectin Binding Assay

A Tris-HCl buffer solution (0.1 M, pH 7.5, including 1 mM MnCl_2_, 1 mM CaCl_2_, and 10 mM NaCl) of FITC-labeled lectin (final conc. of Con A; 2 μM, PNA; 8 μM) was added to a copolymer buffer solution. The resulting solution was incubated at room temperature for 9 h in the dark. After centrifugation, the fluorescence intensity of the supernatant was analyzed by a fluorescence spectrophotometer (*λ*_ex_; 495 nm, *λ*_em_; 518 nm). The association constant was calculated using the Steck–Wallack equation [34].

### 2.11. Cytotoxicity Assessment

Mouse mesenchymal stem cell line (D1 cell, ATCC^®^ Number: CRL-12424™, cell passage 4) was seeded at a cell density of 9000 cells/cm^2^ with the control medium (Dulbecco’s modified eagle’s medium with 10% fetal bovine serum and 1% antibiotics) and cultured overnight to attach the cells. The cells were treated with the control medium (negative control group) or the media with poly(MalVE-*co*-EtMI) and poly(MalMI-*co*-LacVE) separately for an additional four days. The concentrations of each polymer varied from 0 to 100 μg·mL^−1^. WST-8 (Cell Counting Kit, Donjindo, Kumamoto, Japan) was used to assess the cell cytotoxicity in accordance with the manufacturer’s instruction. The absorbance was measured with microplate reader (SpectraMax M5; Molecular Devices, San Jose, CA, USA) at a wavelength of 450 nm. BellCurve (Social Survey Research Information Co., Ltd., Tokyo, Japan) was used for the statistical analysis. Statistical significance was evaluated using a one-way analysis of variance (ANOVA), followed by a Tukey–Kramer test. All results are presented as the mean ± standard deviation (SD) (*n* = 5).

## 3. Results and Discussion

### 3.1. Comparison of Copolymerization of MalVE and EtMI with and without RAFT Agent

Synthesis of 3-[2-(2-vinyloxyethoxy)-ethoxy]-propyne (VEEP) was carried out by a reaction of 2-(vinyloxyethoxy)-ethanol with propargyl bromide in the presence of KOH in DMSO at room temperature for 40 h [33]. Maltosyl azide and lactosyl azide were prepared according to the literature [27,28]. Maltose- or lactose-substituted vinyl ethers (MalVE and LacVE) were synthesized by CuAAC reactions of an alkyne-containing VE (VEEP) with the corresponding disaccharide azides. Maltose-substituted maleimide (MalMI) was synthesized in a similar way to the VE versions instead of using *N*-propargyl MI. The formation of these disaccharide-substituted vinyl monomers was confirmed by ^1^H and ^13^C NMR spectroscopy (Appendix A).

Copolymerization of MalVE and EtMI (feed ratio: MalVE:EtMI = 1:1) was performed in H_2_O/acetonitrile (4/3, *v*/*v*) at 60 °C with 2,2′-azobis-[2-(2-imidazolin-2-yl)propane] dihydrochloride (VA-044) in the presence or absence of BTSE as the RAFT agent (a chain transfer agent widely used in RAFT polymerization [13,14,15]) ([MalVE]_0_/[EtMI]_0_/[VA-044]_0_/[BTSE]_0_ = 50/50/1/1 or 50/50/1/0, [MalVE]_0_ + [EtMI]_0_ = 15 wt %). Size exclusion chromatography (SEC) analysis showed both the polymers obtained with or without BTSE were unimodal but clearly different in molecular weight (MW) and molecular weight distribution (MWD). The polymer obtained with BTSE possesses lower MW and narrower MWD compared that without BTSE (polymer (with BTSE): *M*_n_ = 3900, *M*_w_ = 5800, *M*_w_/*M*_n_ = 1.51; polymer (without BTSE): *M*_n_ = 13,000, *M*_w_ = 21,000, *M*_w_/*M*_n_ = 2.15) (Appendix A). These results suggest the possibility of controlled copolymerization by RAFT process for the copolymerization of MalVE and EtMI with BTSE.

### 3.2. RAFT Copolymerization of MalVE and EtMI

Considering the result of the preliminary experiment, RAFT copolymerization of MalVE and EtMI with BTSE was investigated in detail ([MalVE]_0_/[EtMI]_0_/[VA-044]_0_/[BTSE]_0_ = 50/50/1/1, [MalVE]_0_ + [EtMI]_0_ = 15 wt %). As shown in Figure 2, the copolymerization smoothly proceeded and completed within 2 h. Because only EtMI is capable of homopolymerizing in the RAFT-based radical polymerization process, the conversion of MalVE reached saturation at around 70% while quantitative consumption of EtMI was observed. SEC traces of the obtained copolymers were unimodal and shifted to higher molecular region as the polymerization proceeded with keeping *M*_w_/*M*_n_ values of ca. 1.5 or below (Figure 3a,b). Figure 3b shows the *M*_n_ and *M*_w_/*M*_n_ of the resultant copolymers obtained from the RAFT copolymerization of MalVE and EtMI at various conversions. The observed *M*_n_ values (the filled circles in Figure 3b) determined by the peak intensity ratio of the terminal phenyl protons and protons of MalVE and EtMI repeating units (Figure 4) were found to increase in direct proportion to the monomer conversion and were also in good agreement with the calculated values (the solid line in Figure 3b), supporting the chain lengths of the resultant copolymers poly(MalVE-*co*-EtMI) were well controlled by the RAFT copolymerization. The *M*_n_ values of the obtained copolymers (the filled squares in Figure 3b), which were measured by poly(ethylene glycol) (PEG)-calibrated SEC, were much smaller than the calculated values, which is due to the significant difference in hydrodynamic volume of the bulky disaccharide-carrying glycopolymers with respect to the PEG standards. Figure 4 shows the ^1^H NMR spectrum of poly(MalVE-*co*-EtMI), where all key signals assignable to the maltose moiety (the protons at C-1 and C-1′) and linkage triazole protons of MalVE units, methyl protons of EtMI units, and the terminal phenyl protons at *α*-end from BTSE were clearly observed. The composition ratio and degree of polymerization (*DP*_n_) of poly(MalVE-*co*-EtMI) were estimated using Equations (1)–(5):Composition ratio of MalVE (%) = [*A*_Ha_/(*A*_Ha_ + *A*_Hb_/3)] × 100%(1)
Composition ratio of EtMI (%) = [(*A*_Hb_/3)/(*A*_Ha_ + *A*_Hb_/3)] × 100%(2)
*DP*_n_ of MalVE = [*A*_Ha_/(*A*_Ph_/5)](3)
*DP*_n_ of EtMI = [(*A*_Hb_/3]/(*A*_Ph_/5)](4)
*M*_n NMR_ = 244.4 + 537.5 × (*DP*_n_ of MalVE) + 125.1 × (*DP*_n_ of EtMI)(5)
where *A*_Ha_, *A*_Hb_ and *A*_Ph_ represent the relative peak areas of the triazole proton (peak H_a_), the methylene protons (peak H_b_) and aromatic protons (peak Ph), respectively. Results show that the composition ratio was calculated to be MalVE:EtMI = 42:58 for the copolymer obtained for 120 min reaction. Results of the RAFT copolymerization at MalVE:EtMI = 1:1 feed ratio are summarized in Table 1. An equimolar mixture of MalVE and EtMI was polymerized but the obtained copolymers were found to be almost constant in composition ratio of MalVE/EtMI ≈ 40/60 irrespective of the monomer conversion. This is because only EtMI possesses homopolymerization ability. As a result, the RAFT copolymerization in this study afforded a novel glycopolymer having nearly periodic structure, where major parts possess alternating structure (-(MalVE-EtMI)_n_-) accompanied by a small part of consecutive sequence of EtMI (–EtMI-EtMI-). The predicted structure was supported by MALDI-TOF-MS analysis in the presence of 2,5-dihydroxybenzoic acid (2,5-DHB) as the matrix and NaCl as the ionizing agent (Figure 5). The spectrum shows a dominant series of the alternating sequence (-(MalVE-EtMI)_n_-) in which each peak is separated by intervals of 125 Da for EtMI and 537 Da for MalVE, accompanied by a minor series of the consecutive sequence of EtMI ((–EtMI-EtMI-)). Based on the composition ratio of the copolymers (MalVE/EtMI ≈ 40/60) determined by ^1^H NMR analysis (Table 1), the content of the alternating sequences (-(MalVE-EtMI)_n_-) in one polymer chain is evaluated to be ca. 80% on average. To improve the extent of control in the alternating structure of the copolymers, copolymerization was examined by varying the initial monomer feed ratio [MalVE]_0_/([MalVE]_0_ + [EtMI]_0_) in the range from 0.1 to 0.85. As shown in Figure 6, the copolymer composition ratios slightly changed depending on the monomer feed ratio, consequently almost alternating structure was attained using an excess amount of MalVE ([MalVE]_0_/([MalVE]_0_ + [EtMI]_0_) = 0.85) for copolymerization. The monomer reactivity ratio was fitted by a penultimate model, as shown in Figure 6 [24]. In this model, there are four parameters for the monomer reactivity ratios, *r*_11_, *r*_12_, *r*_21_, and *r*_22_ (Appendix A). Considering the consecutive addition of VE moiety hardly occurs under the radical polymerization condition, we assumed that *r*_11_ and *r*_21_ were zero, the *r*_12_ and *r*_22_ were thus calculated to be 0.90 and 0.20, respectively, by the Kelen–Tüdõs method for the penultimate model. In Figure 6, the plateau region above 0.67 of MalVE content in feed indicated the glycopolymers (poly(MalVE-*co*-EtMI)) consisting of almost alternating structure.

### 3.3. RAFT Copolymerization of LacVE and EtMI

RAFT copolymerization of LacVE and EtMI was also investigated under similar conditions as those of MalVE and EtMI ([LacVE]_0_/[EtMI]_0_/[VA-044]_0_/[BTSE]_0_ = 50/50/1/1, [LacVE]_0_ + [EtMI]_0_ = 15 wt %). The RAFT copolymerization proceeded smoothly without an induction period, and EtMI was quantitatively consumed and the conversion of LacVE reached over 85% within 80 min (Appendix A). The results of the RAFT copolymerization of LacVE and EtMI are summarized in Table 2. Throughout the polymerization, all SEC chromatograms of the obtained copolymers were unimodal and their polydispersity indices were maintained at around 1.5 or below (Appendix A). As in the case of the combination of MalVE and EtMI, due to the homopolymerizability of EtMI, the copolymerization at LacVE:EtMI = 1:1 feed ratio results in the formation of copolymers being almost constant in composition ratio of LacVE/EtMI ≈ 40/60 irrespective of the monomer conversion. These results indicate the BTSE-mediated copolymerization of LacVE and EtMI proceeded in a controlled fashion.

### 3.4. RAFT Copolymerization of LacVE and MalMI

The goal of our research was the synthesis of alternating glycopolymers as glycosaminoglycan mimics, where two kinds of carbohydrate-substituted vinyl monomers are alternately connected with each other. In this study, we developed a new synthetic approach for the synthesis of alternating glycopolymers with controlled molecular weight and polymer chain length distribution. Based on the results of the synthesis of periodic glycopolymers mentioned above, we here demonstrated the RAFT copolymerization of two kinds of disaccharide-substituted vinyl monomers, LacVE and MalMI. RAFT copolymerization of LacVE and MalMI was carried out using BTSE and VA-044 in H_2_O/acetonitrile (4/3, *v*/*v*) at 60 °C ([LacVE]_0_/[MalMI]_0_/[VA-044]_0_/[BTSE]_0_ = 50/50/1/1, [LacVE]_0_ + [MalMI]_0_ = 10 wt %). The copolymerization smoothly occurred and the conversions of LacVE and MalMI reached 67% and 81%, respectively, after a period of 4 h (Appendix A). Unlike the complete consumption in the copolymerization using EtMI as a comonomer, MalMI was not quantitatively consumed, probably due to the steric hindrance of the disaccharide-substituted pendant structure. The polymerization results are summarized in Table 3. SEC analysis indicated all the chromatograms of the copolymers were unimodal (Appendix A) and their *M*_w_/*M*_n_ values were around 1.5 or below. Figure 7 depicts ^1^H MMR spectrum in D_2_O of the obtained copolymer poly(LacVE-*co*-MalMI). Along with the linkage triazole proton (Peaks a and d), two pairs of characteristic proton signals at C-1 and C-1′ of the disaccharide moieties (Peaks e and f for MalMI unit, and Peaks b and c for LacVE unit) were clearly observed. Moreover, aromatic protons assignable to the benzyl moiety at *α*-end, which originated from BTSE, were observed. Based on the integrated intensities of the key signals, the composition ratio was calculated to be LacVE/MalMI ≈ 36/64. All the copolymers obtained at various conversions possessed similar compositions. In comparison with the composition ratios of poly(MalVE-*co*-EtMI) and poly(LacVE-*co*-EtMI) (VE/MI ≈ 40/60), the ratios are not significantly different from that of poly(LacVE-*co*-MalMI), showing the sequence regulation in the copolymer synthesis by RAFT copolymerization may be affected only by the reactivity the vinyl moieties not by the steric hindrance of the pendant carbohydrate residues. Therefore, the target alternating glycopolymers are expected to be synthesized by the RAFT copolymerization of an excess use of carbohydrate-substituted VE.

### 3.5. Lectin Binding Assay

The binding property of copolymers poly(MalVE-*co*-EtMI) and poly(LacVE-*co*-EtMI) were investigated, and the association constants (*K*_a_) for the lectin–saccharide interaction were estimated using fluorescein-labeled lectins (Figure 8). The addition of copolymer to the buffer solution of FITC-labeled lectin resulted in decreasing the intensity of fluorescence due to multivalent binding of *α*-glucoside residue in maltose with Con A and of *β*-galactoside residue in lactose with PNA, respectively. No glycopolymer–lectin binding was observed when non-corresponding lectin was added to each copolymer solution in the turbidity test (Appendix A). The values of *K*_a_ were 1.3 × 10^5^ M^−1^ (poly(MalVE-*co*-EtMI) and Con A) and 5.0 × 10^4^ M^−1^ (poly(LacVE-*co*-EtMI) and PNA), respectively. The *K*_a_ value of free saccharide with corresponding lectin is reported in the order of 10^3^ M^−1^ [35]. Therefore, these results indicate that the periodic glycopolymers poly(MalVE-*co*-EtMI) and poly(LacVE-*co*-EtMI) strongly and specifically interacted with corresponding lectin in aqueous condition due to the multivalency of saccharide in glycopolymers.

### 3.6. In Vitro Cytotoxicity

Cytotoxicity studies (Figure 9) show that there was no significant difference in cellular viability when D1 cells were treated with the media including poly(MalVE-*co*-EtMI) or poly(LacVE-*co*-MalMI) in comparison with that of medium without polymers. The results indicate that both polymers may potentially cause no or poor cytotoxicity for the cells, when concentrations are below 100 μg mL^−1^.

## 4. Conclusions

In this study, we demonstrated that the RAFT copolymerization of a combination of carbohydrate-substituted VE (as an electron-rich vinyl monomer) and EtMI (as an electron-deficient vinyl monomer) is a promising synthetic approach for yielding novel type of periodic glycopolymers. Moreover, almost alternating sequence control in copolymers can be achieved by regulating the initial monomer feed ratio. It should be emphasized that alternating glycopolymers as glycosaminoglycan mimics, in which two kinds of saccharide moieties are alternately incorporated in every pendant, can be designed by the use of a combination of carbohydrate-containing VE and MI monomers. Another possible advantage of the present synthetic strategy is that it would be capable of affording mimics of non-naturally occurring polysaccharides with alternating structure, which may possess novel physiological and/or pharmaceutical functions. RAFT copolymerization of various combinations of carbohydrate-substituted VE and MI derivatives are in progress and will be reported elsewhere.

## Figures and Tables

**Figure 1 polymers-11-00070-f001:**
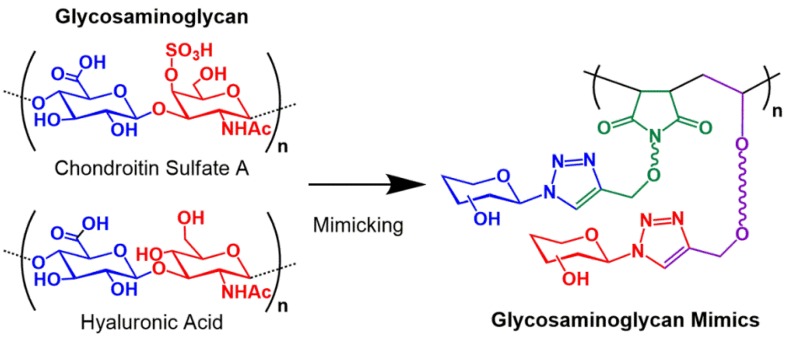
Concept of glycosaminoglycan mimicking in this study.

**Figure 2 polymers-11-00070-f002:**
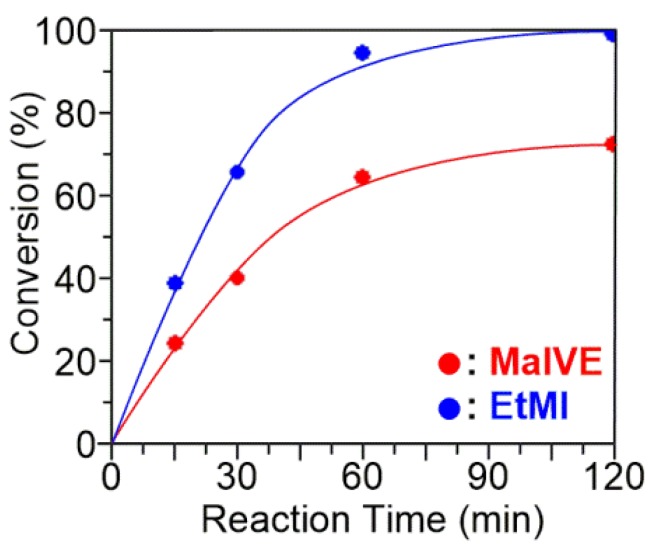
Time–monomer conversion curves for the RAFT copolymerization of MalVE and EtMI with BTSE.

**Figure 3 polymers-11-00070-f003:**
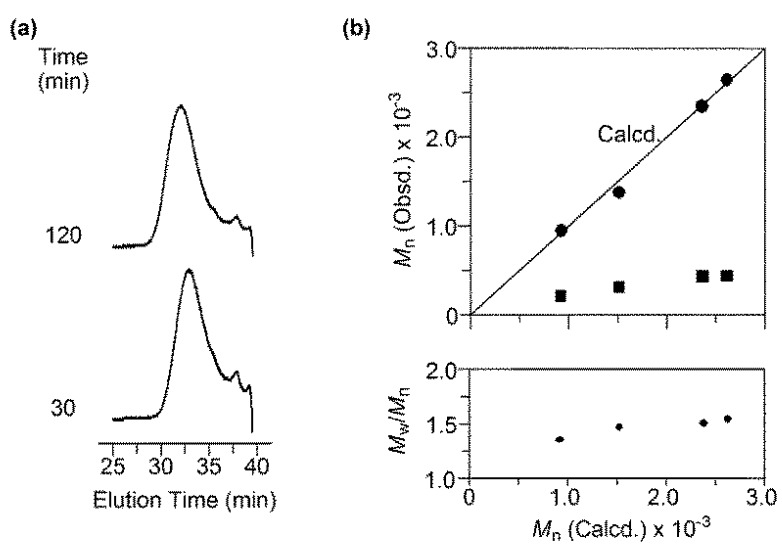
(**a**) SEC curves of poly(MalVE-*co*-EtMI) using 0.2 mol·L^−1^ NaNO_3_ aq. as the eluent; and (**b**) experimentally observed *M*_n_ and *M*_w_/*M*_n_ value of poly(MalVE-*co*-EtMI) plotted against theoretical *M*_n_ of poly(MalVE-*co*-EtMI). Filled circles and squares correspond to the *M*_n_ data obtained by ^1^H NMR and SEC, respectively.

**Figure 4 polymers-11-00070-f004:**
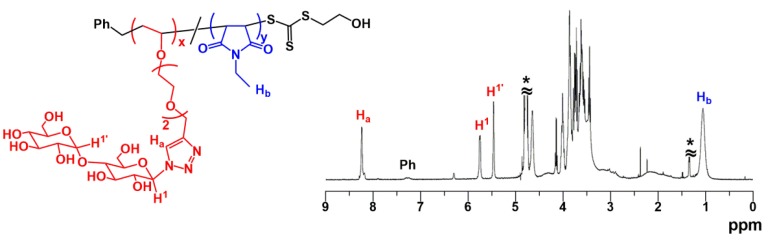
^1^H NMR spectrum of poly(MalVE-*co*-EtMI) in D_2_O.

**Figure 5 polymers-11-00070-f005:**
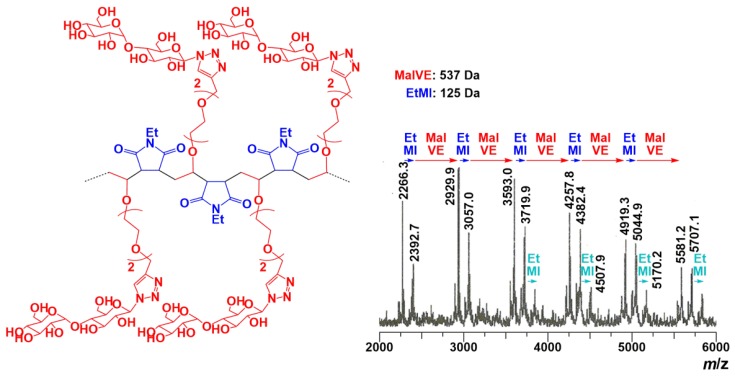
MALDI-TOF-MS spectrum of poly(MalVE-*co*-EtMI).

**Figure 6 polymers-11-00070-f006:**
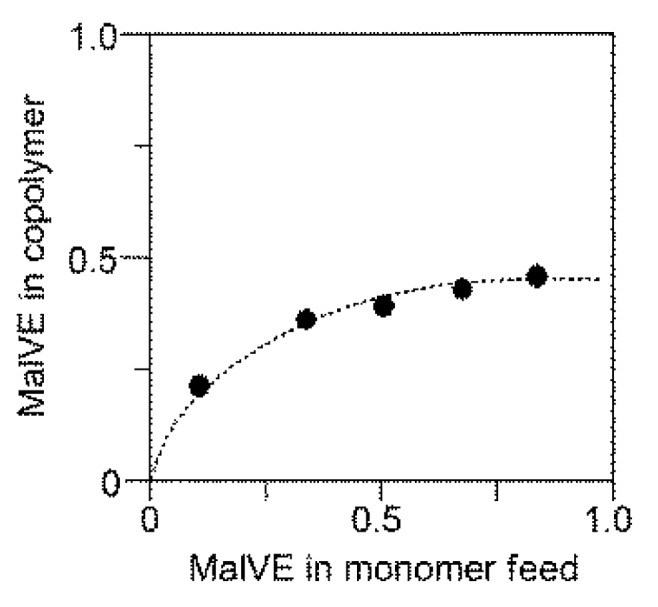
Copolymer composition curve for the copolymerization of MalVE (M_1_) and EtMI (M_2_). The dotted line in plot was fitted by the Kelen–Tüdõs method, assuming that the values of *r*_11_ and *r*_21_ are 0.

**Figure 7 polymers-11-00070-f007:**
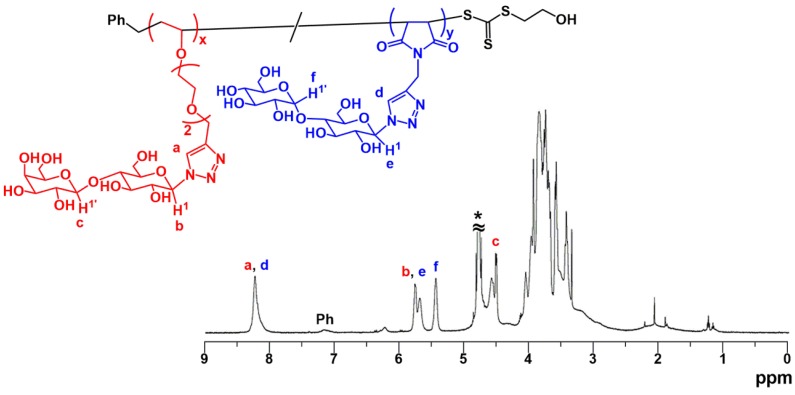
^1^H NMR spectrum of poly(LacVE-*co*-MalMI) in D_2_O.

**Figure 8 polymers-11-00070-f008:**
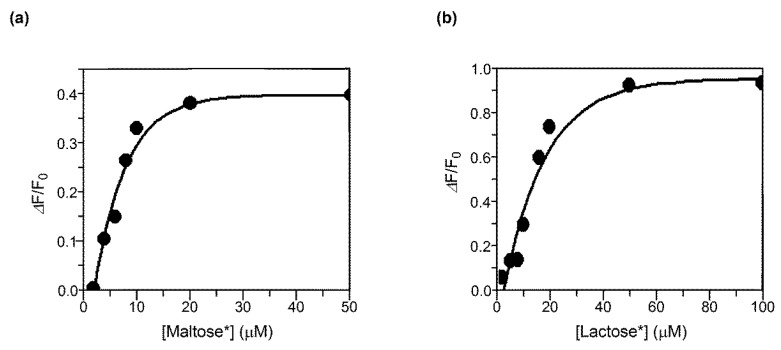
Quenching of the fluorescence intensity of: (**a**) FITC-labeled Con A by the addition of poly(MalVE-*co*-EtMI); and (**b**) FITC-labeled PNA by the addition of poly(LacVE-*co*-EtMI). [Maltose*] and [Lactose*] correspond to the concentration of the pendant maltose moiety in the poly(MalVE-*co*-EtMI) and the pendant lactose moiety in the poly(LacVE-*co*-EtMI), respectively.

**Figure 9 polymers-11-00070-f009:**
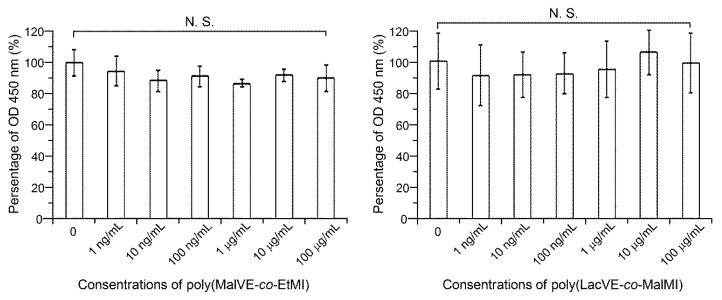
Cytotoxicity studies for the obtained glycopolymers. The mouse mesenchymal stem cell line (D1 cell) was treated with/without poly(MalVE-*co*-EtMI) or poly(LacVE-*co*-MalMI) for four days. The data were normalized to the control group without polymers. The results are presented as percentages (control group: 100%). Data represent the mean with standard deviation (*n* = 5 per group). N.S., not significant.

**Table 1 polymers-11-00070-t001:** RAFT copolymerization of MalVE and EtMI ^1^.

Reaction Time (min)	Conversion (%) ^2^		*M* _n SEC_ ^3^	*M* _n NMR_ ^2^	*M*_w_/*M*_n_^3^	Composition Ratio (%) ^1^	
	MalVE	EtMI				MalVE	EtMI
15	24	39	2100	9600	1.34	41	59
30	40	66	3100	14,000	1.46	42	58
60	65	95	4200	24,000	1.49	40	60
120	73	100	4400	27,000	1.53	42	58

^1^ Polymerization conditions: H_2_O/acetonitrile = 4/3 (*v*/*v*), 60 °C, [MalVE]_0_ + [EtMI]_0_ = 15 wt %, [MalVE]_0_/[EtMI]_0_/[VA-044]_0_/[BTSE]_0_ = 50/50/1/1. ^2^ Determined by ^1^H NMR. ^3^ Estimated by PEG-calibrated SEC.

**Table 2 polymers-11-00070-t002:** RAFT copolymerization of LacVE and EtMI ^1^.

Reaction Time (min)	Conversion (%) ^2^		*M* _n SEC_ ^3^	*M* _n NMR_ ^2^	*M*_w_/*M*_n_^3^	Composition Ratio (%) ^1^	
	LacVE	EtMI				LacVE	EtMI
10	28	37	3600	10,000	1.46	43	57
20	49	66	4600	22,000	1.49	42	58
40	75	92	6200	29,000	1.53	43	57
80	87	100	6600	28,000	1.51	43	57

^1^ Polymerization conditions: H_2_O/acetonitrile = 4/3 (*v*/*v*), 60 °C, [LacVE]_0_ + [EtMI]_0_ = 15 wt %, [LacVE]_0_/[EtMI]_0_/[VA-044]_0_/[BTSE]_0_ = 50/50/1/1. ^2^ Determined by ^1^H NMR. ^3^ Estimated by PEG-calibrated SEC.

**Table 3 polymers-11-00070-t003:** RAFT copolymerization of LacVE and MalMI ^1^.

Reaction Time (min)	Conversion (%) ^2^		*M* _n_ ^3^	*M*_w_/*M*_n_^3^	Composition Ratio (%) ^2^	
	LacVE	MalMI			LacVE	MalMI
20	3	22	4700	1.40	36	64
30	18	34	5000	1.48	35	65
60	30	49	5800	1.52	36	64
120	61	74	5900	1.50	36	64
240	67	81	6000	1.47	36	64

^1^ Polymerization conditions: H_2_O/acetonitrile = 4/3 (*v*/*v*), 60 °C, [LacVE]_0_ + [MalMI]_0_ = 10 wt %, [LacVE]_0_/[MalMI]_0_/[VA-044]_0_/[BTSE]_0_ = 50/50/1/1. ^2^ Determined by ^1^H NMR. ^3^ Estimated by PEG-calibrated SEC.

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
