# Peer review of "The First Synthesis of Periodic and Alternating Glycopolymers by RAFT Polymerization: A Novel Synthetic Pathway for Glycosaminoglycan Mimics"

_polymers, 2019, doi:10.3390/polym11010070_

Round 1

Reviewer 1 Report

The manuscript reports the results from a study on synthesis of glycopolymers by RAFT  copolymerization of  maltose- and lactose-substituted vinyl ethers  with maltose-functionalized maleimide. The obtained glycopolymers present new products with predictable structure and composition. They are non-cytotoxic  and exhibit cluster effect with lectins. In that aspect the study possesses novelty and prospects for further development of the product design.

However, there are still some issues that should be addressed prior acceptance for publication:

Lines 80-81: It is said “To confirm the potential ability of the glycopolymers in this study as promising materials in biomedical and pharmaceutical field, cytotoxicity assessment was provided.” I would like to suggest the “ability” to be replaced with “applicability”, i.e. “To confirm the potential applicability of the glycopolymers in this study as promising materials in biomedical and pharmaceutical field, cytotoxicity assessment was provided.”

Lines 127-130; Lines 141-144 and Lines 153-155 contain description of the 1H and 13C NMR spectra of the obtained functionalized monomers MalVE, LacVE and MalMI. There are ambiguous assignments of the signals in the 1H NMR spectra and no assignment of the signals in 13C NMR spectra. It will be helpful for the readers if monomer structures with labeled atoms are inserted in FigS1 (at least for the 1H NMR spectra) and signals assigned accordingly.

Line 215: It is written in the Results and Discussion section that “Maltosyl azide and lactosyl azide were prepared according to a literature [6] with some modifications”. The modifications are not described in the Materials and Methods section therefore it is recommendable a brief description of the modified procedure to be included, or if published elsewhere to add the reference.

Lines 225-229: “SEC analysis showed both the polymers obtained with or without  BTSE were unimodal but clearly different in molecular weight (MW) and molecular weight distribution (MWD), that is, the polymer obtained with BTSE possesses lower MW and narrower  MWD compared that without BTSE [polymer (with BTSE): Mn = 3,900, Mw = 5,800, Mw/Mn = 1.51; polymer (without BTSE): Mn = 13,000, Mw = 21,000, Mw/Mn = 2.15] (Figure S2). “ The sentence is too long and needs revision.

Lines 229-230: The summarizing sentence “These results indicate that the copolymerization of MalVE and EtMI proceeded in a RAFT polymerization mechanism to give polymers with controlled structure.” should be revised. SEC data provide information about the molecular-mas characteristics and the comparison of the values for Mn and Mw/Mn is not enough to derive conclusion about obtaining a polymers with controlled structure.

Lines 247-249: The measured by SEC smaller Mn values than those obtained by NMR or the calculated ones could be due to the comb-like structure of the disaccharide-carrying glycopolymers with respect to the linear PEG standards.

In my opinion there is a need of explanation of the observed difference between the monomers conversion and the composition of the obtained copolymer derived from equimolar LacVE and MalMI  feeding mixture (Table 3) and both measured by means of 1H NMR spectroscopy.

In Scheme S1 the lower left arrow is pointing to missing object.

English language and style are of good quality; however, a careful check will improve the quality of the manuscript, with a special note to the Conclusions.

Author Response

We appreciate your valuable comments and suggestions. With careful consideration of your comments, we have modified the manuscript and answered to the questions as follows.

Lines 80-81: It is said “To confirm the potential ability of the glycopolymers in this study as promising materials in biomedical and pharmaceutical field, cytotoxicity assessment was provided.” I would like to suggest the “ability” to be replaced with “applicability”, i.e. “To confirm the potential applicability of the glycopolymers in this study as promising materials in biomedical and pharmaceutical field, cytotoxicity assessment was provided.”

--> According to your suggestion, we have revised the sentences as follows;

Page 2, lines 80-82;

“To confirm the potential applicability of the glycopolymers in this study as promising materials in biomedical and pharmaceutical field, cytotoxicity assessment was provided”

Lines 127-130; Lines 141-144 and Lines 153-155 contain description of the 1H and 13C NMR spectra of the obtained functionalized monomers MalVE, LacVE and MalMI. There are ambiguous assignments of the signals in the 1H NMR spectra and no assignment of the signals in 13C NMR spectra. It will be helpful for the readers if monomer structures with labeled atoms are inserted in FigS1 (at least for the 1H NMR spectra) and signals assigned accordingly.

--> According to your suggestions, we have added the monomer structures with labeled atoms and assignment of the signals in Figure S1

Line 215: It is written in the Results and Discussion section that “Maltosyl azide and lactosyl azide were prepared according to a literature [6] with some modifications”. The modifications are not described in the Materials and Methods section therefore it is recommendable a brief description of the modified procedure to be included, or if published elsewhere to add the reference.

--> According to your suggestion, we have revised the sentence and added another reference which describes the disaccharide azides preparation to the manuscript.

“Maltosyl azide and lactosyl azide were prepared according to literatures [6].”

As ref. 6b; (b) Tanaka, T.; Ishitani, H.; Miura, Y.; Oishi, K.; Takahashi, T.; Suzuki, T.; Shoda, S.; Kimura, Y. Protecting-Group-Free Synthesis of Glycopolymers Bearing Sialyloligosaccharide and Their High Binding with the Influenza Virus. ACS Macro Lett. 2014, 3, 1074–1078.

Lines 225-229: “SEC analysis showed both the polymers obtained with or without  BTSE were unimodal but clearly different in molecular weight (MW) and molecular weight distribution (MWD), that is, the polymer obtained with BTSE possesses lower MW and narrower  MWD compared that without BTSE [polymer (with BTSE): Mn = 3,900, Mw = 5,800, Mw/Mn = 1.51; polymer (without BTSE): Mn = 13,000, Mw = 21,000, Mw/Mn = 2.15] (Figure S2). “ The sentence is too long and needs revision.

--> According to your suggestion, we have revised the sentence as follows;

“SEC analysis showed both the polymers obtained with or without BTSE were unimodal but clearly different in molecular weight (MW) and molecular weight distribution (MWD). The polymer obtained with BTSE possesses lower MW and narrower MWD compared that without BTSE [polymer (with BTSE): Mn = 3,900, Mw = 5,800, Mw/Mn = 1.51; polymer (without BTSE): Mn = 13,000, Mw = 21,000, Mw/Mn = 2.15] (Figure S2).”

Lines 229-230: The summarizing sentence “These results indicate that the copolymerization of MalVE and EtMI proceeded in a RAFT polymerization mechanism to give polymers with controlled structure.” should be revised. SEC data provide information about the molecular-mas characteristics and the comparison of the values for Mn and Mw/Mn is not enough to derive conclusion about obtaining a polymers with controlled structure.

--> As you pointed out, the comparison of the values for Mn and Mw/Mn (the description in the text and Figure S2) is not enough to conclude that the polymer product obtained under the RAFT polymerization possesses controlled structure. That is confirmed in the next section 3.2. We have revised the sentence as follows;

“These results suggest that the possibility of controlled copolymerization by RAFT process for the copolymerization of MalVE and EtMI with BTSE.

Lines 247-249: The measured by SEC smaller Mn values than those obtained by NMR or the calculated ones could be due to the comb-like structure of the disaccharide-carrying glycopolymers with respect to the linear PEG standards.

In my opinion there is a need of explanation of the observed difference between the monomers conversion and the composition of the obtained copolymer derived from equimolar LacVE and MalMI feeding mixture (Table 3) and both measured by means of 1H NMR spectroscopy.

--> We appreciate your valuable comments and suggestions. As you commented, the difference between the Mn values by PEG-calibrated SEC and those by NMR is surely due to the comb-like structure of poly(MalVE-co-EtMI) with respect to the linear PEGs, causing the significant difference in hydrodynamic volume.

      Because the proton peaks of the phenyl group at alpha (a)–terminus of poly(LacVE-co-MalMI) were broader than those of the poly(MalVE-co-EtMI), we recognize that they can be utilized for determination of the composition ratios of LacVE and MalMI units in copolymers, but may be less reliable to evaluate the degree of polymerization of LacVE and MalMI units by the peak intensity ratio of the alpha (a)–end and pendant functions in the 1H NMR spectra. We are wondering why such difference in signal’s intensity and shape was observed for poly(LacVE-co-MalMI). Possible reasons are the enhanced hydrophilicity and different conformation of poly(LacVE-co-MalMI) having disaccharide residues in every pendants, thereby the hydrophobic phenyl group at alpha (a)–end may have more restricted mobility. Thus, we are going to estimate the absolute MWs of these glycopolymers by the SEC system equipped with a light scattering detector.

      As for the difference in monomer conversions between the pairs of MalVE/EtMI and LacVE/EtMI and that of LacVE/MalMI, the incomplete consumption of MalMI in the LacVE/MalMI copolymerization is probably due to the steric hindrance of the combined use of disaccharide-substituted VE and MI. This has already been described in the original text in lines 326-329. However, as described in the original text (lines 337-341), the sequence regulation in the copolymer synthesis by RAFT copolymerization may be affected only by the reactivity the vinyl moieties not by the steric hindrance of the monomers.

In Scheme S1 the lower left arrow is pointing to missing object.

--> According to your suggestion, we have added the molecular structure in Scheme S1.

Reviewer 2 Report

In this contribution, Minoda and his coworkers reported on RAFT copolymerization of lactose-substituted VE (LacVE) and maltose-substituted MI (MalMI) to obtain glycopolymers periodically carrying two kinds of disaccharide residues in the pendants, on the basis of investigating RAFT copolymerization of maltose-substituted VE (MalVE) and ethyl maleimide (EtMI). The chemical structures were characterized by 1H NMR, GPC and MALDI-TOF, as well. Moreover, The glycocluster effect and cell-cytotoxicity of the resultant glycopolymers poly(MalVE-co-EtMI) and poly(LacVE-co-EtMI) were further assessed. However, in the aspect of the chemical structure, it seems insufficiency to support the periodic structures of copolymers in the current results. Only MALDI-TOF of poly(MalVE-co-EtMI) was shown in this manuscript, but was short of that of poly(LacVE-co-MalMI) to directly substantiate the periodical structure. There was a great disparity in molecular weights between obtained by 1H NMR and MALDI-TOF, and it needed to give reasonable explanation.

In addition, several suggestions could be considered.

In the Tables, please give the necessary polymerization condition including feeding ratio, temperature and monomer type, et.al. in the footnote

If possible, please add the molecular weights of each peak in MALDI-TOF.

Please give detailed calculation information of molecular weights and composition ratios of the resultant polymers from 1H NMR.    

Author Response

We appreciate your valuable comments and suggestions. With careful consideration of your comments, we have modified the manuscript and answered to the questions as follows.

However, in the aspect of the chemical structure, it seems insufficiency to support the periodic structures of copolymers in the current results. Only MALDI-TOF of poly(MalVE-co-EtMI) was shown in this manuscript, but was short of that of poly(LacVE-co-MalMI) to directly substantiate the periodical structure. There was a great disparity in molecular weights between obtained by 1H NMR and MALDI-TOF, and it needed to give reasonable explanation.

--> As described in the text, the sequence structure of poly(MalVE-co-EtMI) has been verified by both MALDI-TO-MS (Figure 5) and investigation on the copolymerization based on a penultimate model (Figure 6 and Scheme S1). As the results, we confirmed that MalVE : EtMI = 1 : 1 feed ratio yielded periodic poly(MalVE-co-EtMI), where the term “periodic” means the resultant copolymers is not alternating one because it consists of ca. 80% of alternating sequences [-(MalVE-EtMI)n-] and ca. 20% of consecutive sequences of EtMI (–EtMI-EtMI-). However, as you could see, the use of an excess feed of MalVE resulted in the successful synthesis of almost alternating poly(MalVE-co-EtMI) having quite a few amount of -EtMI-EtMI- sequences. This was supported by the plateau region at the MalVE content in feed above 0.67 in the copolymer composition curve based on Kelen-Tüdõs method (Figure 6) (Refs. 5a-h, 7a-b). Thus, we added the following sentence;

      “In Figure 6, the plateau region above 0.67 of MalVE content in feed indicated the glycopolymers [poly(MalVE-co-EtMI)] consisting of almost alternating structure.”

      As you mentioned, we recognize that further experimental results are indeed necessary to clarify the detailed monomer sequence of poly(LacVE-co-MalMI) having different two pendant disaccharides. However, based on the results of the copolymerization of MalVE and EtMI, it is surely expected that the use of an excess feed of LacVE would form almost alternating poly(LacVE-co-MalMI). Due to the small difference in molecular weight of the disaccharide-substituted monomers, LacVE and MalMI, we think it is rather difficult to analyze the MALDI-TOF-MS spectrum of the alternating poly(LacVE-co-MalMI). Now, we are pursuing the synthesis of an alternating copolymer from a monosaccharide-appended MI and a disaccharide-appended VE, and the results will be reported elsewhere.

      As for your comment in the last sentence, because of the difficulty in ionization of relatively high molecular weight poly(MalVE-co-EtMI) (> ca. 7000 Da), we analyzed the sequence structure by MALDI-TOF-MS by focusing on the peaks of the m/z range of 2000-6000 Da. It would be reasonably acceptable that the present MALDI-TOF-MS data supports the formation of the copolymers with similar sequence structure irrespective of the molecular weight because the copolymerization (based on the sequence of the two monomer insertions) similarly proceeded throughout the polymerization (Figure 2 and Table 1).

In the Tables, please give the necessary polymerization condition including feeding ratio, temperature and monomer type, et.al. in the footnote

--> According to your suggestion, we have added the polymerization conditions to the footnotes in all Tables:

Table 1; “Polymerization conditions: H2O / acetonitrile = 4 /3 (v/v), 60 °C, [MalVE]0 + [EtMI]0 = 15 wt%, [MalVE]0 / [EtMI]0 /[VA-044]0 / [BTSE]0 = 50 / 50 / 1 / 1.”

Table 2; “Polymerization conditions: H2O / acetonitrile = 4 /3 (v/v), 60 °C, [LacVE]0 + [EtMI]0 = 15 wt%, [LacVE]0 / [EtMI]0 /[VA-044]0 / [BTSE]0 = 50 / 50 / 1 / 1.”

Table 3; “Polymerization conditions: H2O / acetonitrile = 4 /3 (v/v), 60 °C, [LacVE]0 + [MalMI]0 = 10 wt%, [LacVE]0 / [MalMI]0 /[VA-044]0 / [BTSE]0 = 50 / 50 / 1 / 1.”

If possible, please add the molecular weights of each peak in MALDI-TOF

--> According to your suggestion, we have added the molecular weights of each peak improved Figure 5.

Please give detailed calculation information of molecular weights and composition ratios of the resultant polymers from 1H NMR.

--> According to your suggestion, we have added the calculation information of molecular weights and composition ratios of the resultant polymers from 1H NMR. We thus added the equations employed for calculating those values:

The composition ratio and degree of polymerization (DPn) of poly(MalVE-co-EtMI) were estimated using the following equations (1)-(5):

Composition ratio of MalVE (%) = [AHa/(AHa+AHb/3)]   x 100%

Composition ratio of EtMI (%) = [(AHb/3)/(AHa+AHb/3)] x 100%

DPn of MalVE = [AHa/( APh/5)]

DPn of EtMI = [(AHb/3]/( APh/5)]

Mn NMR = 244.4 + 537.5 x (DPn of MalVE) + 125.1 x (DPn of EtMI)

(1)

(2)

(3)

(4)

(5)

where AHa, AHb and APh represent the relative peak areas of the triazole proton (peak Ha), the methylene protons (peak Hb) and aromatic protons (peak Ph), respectively.”

Reviewer 3 Report

In this manuscript, Minoda and coworkers developed a novel approach to periodic and alternating glycopolymers by harnessing RAFT polymerization technique. In their research, novel design and synthesis of various disaccharide vinyl ether monomers (electronic rich) and disaccharide-substituted maleimide monomers (electronic deficient) was achieved. Furthermore, those MI monomers and VE monomers were copolymerized, leading to alternating composition of MI and VE. Finally, the periodic glycopolymer was examined for lectin binding assay and cytotoxicity study. Overall, the design and experiments are solid. Their results are in good agreement with their conclusions. Considering that this is the first example of making alternating glycopolymers by RAFT, the novelty of this research is high and I trust it will attract wide audience in polymer chemistry. Therefore, I would recommend acceptance of this manuscript to Polymers. However, the authors should address the comments below to further improve the manuscript.

Comments

1. For RAFT copolymerization of MalVE and EtMI, the ratio of VA-044 to CTA was 1/1. This loading of initiator is perhaps too high since the PDI of the resulting polymer is 1.5, which is a little high for RAFT systems. Did the authors try to use a lower loading of VA-044 such as 0.2 or 0.1 equivalence of 1.0 equivalent CTA?

2. Did the authors consider using styrenic monomers for alternating polymerization with MI monomer? In general, styrenic monomers are more rich in electrons and can lead to a higher alternating preference in copolymerization with MI.

3. The MW from GPC is much lower than theoretical. To get a better idea of MW of the polymers, did the authors consider using Multi-angel light scattering detector in combination with GPC?

4. The references in [3] for controlled radical polymerization can be updated with most recent important reviews and papers. Highly suggest adding Sumerlin et al. Prog. Polym. Sci., 2018, https://doi.org/10.1016/j.progpolymsci.2018.09.006. And Sumerlin et al. Nature Chemistry, 2017, 9, 817-823.

5. In table 3, please also add a column of Mn,NMR to be consistent with other tables.

Minors

1. Please define SEC throughout the manuscript.

2. In page 5, line 218, “azide” should be “azides”.

3. In page 5, line 219, add “of” after “instead”.

4. In page 7, caption of figure 2, add “monomer” before “conversion”.

Author Response

We appreciate your valuable comments and suggestions. With careful consideration of your comments, we have modified the manuscript and answered to the questions as follows.

1. For RAFT copolymerization of MalVE and EtMI, the ratio of VA-044 to CTA was 1/1. This loading of initiator is perhaps too high since the PDI of the resulting polymer is 1.5, which is a little high for RAFT systems. Did the authors try to use a lower loading of VA-044 such as 0.2 or 0.1 equivalence of 1.0 equivalent CTA?

--> Thank you for your valuable comments and suggestions. As you pointed out, we also recognize that further experiments are necessary for optimizing the present RAFT copolymerization of carbohydrate-substituted VEs and MIs. We have already attempted the RAFT copolymerization of MalMI and an OH-functionalized VE with 0.5 equivalence of VA-044 to the amount of CTA, however, the PDIs of the obtained copolymers still remained being around 1.5. Following your suggestion, we are going to perform the RAFT copolymerization of these monomers with varying the ratio of VA-044 to CTA in a wider range. We believe that the results will be reported elsewhere.

2. Did the authors consider using styrenic monomers for alternating polymerization with MI monomer? In general, styrenic monomers are more rich in electrons and can lead to a higher alternating preference in copolymerization with MI.

--> We appreciate your nice suggestion. Though not described in the original manuscript, we have already investigated about the RAFT copolymerization of MI and maltose-appended styrene derivative (MalSt). Contrary to our expectation as well as yours, the synthesis of the alternating glycopolymers from MI and MalSt was somewhat complicated but eventually achieved by optimizing the reaction conditions. We are planning to follow up this work and to publish in the near future.

3. The MW from GPC is much lower than theoretical. To get a better idea of MW of the polymers, did the authors consider using Multi-angle light scattering detector in combination with GPC?

--> As you pointed out, evaluation of the absolute MWs of glycopolymers is one of the important issues to be considered because their MWs are usually underestimated due to the rather small hydrodynamic volume relative to MW. We are going to estimate the absolute MWs of our glycopolymers by the SEC system equipped with a light scattering detector in the continuing work.

4. The references in [3] for controlled radical polymerization can be updated with most recent important reviews and papers. Highly suggest adding Sumerlin et al. Prog. Polym. Sci., 2018, https://doi.org/10.1016/j.progpolymsci.2018.09.006. And Sumerlin et al. Nature Chemistry, 2017, 9, 817-823.

--> We appreciate the useful information. Following your suggestion, we added those references you kindly cited to the revised manuscript.

As ref. 3f; Sun, H.; Kabb, C.P.; Dai, Y.; Hill, M.R.; Ghiviriga, I.; Bapat, A.P.; Sumerlin, B.S. Macromolecular metamorphosis via stimulus-induced transformations of polymer architecture. Nature Chemistry 2017, 9, 817-823.

As ref. 3g; Sun, H.; Kabb, C.P.; Sims, M.B.; Sumerlin, B.S. Architecture-transformable polymers: Reshaping the future of stimuli-responsive polymers. Prog. Polym. Sci., 2018, in press. DOI: 10.1016/j.progpolymsci.2018.09.006.

5. In table 3, please also add a column of Mn,NMR to be consistent with other tables.

--> We appreciate your valuable suggestion. Because the proton peaks of the phenyl group at alpha (a)–terminus of poly(LacVE-co-MalMI) were broader than those of the poly(MalVE-co-EtMI), we recognize it may be less reliable to evaluate the degree of polymerization of LacVE and MalMI units by the peak intensity ratio of the alpha (a)–end and pendant functions in the 1H NMR spectra. We are wondering why such difference in signal’s intensity and shape was observed for poly(LacVE-co-MalMI). One possible reason is the enhanced hydrophilicity of poly(LacVE-co-MalMI) having disaccharide residues in every pendants, thereby the hydrophobic phenyl group at alpha (a)–end may have more restricted mobility. As mentioned above, we will try to estimate the absolute MWs of these glycopolymers by the SEC system equipped with a light scattering detector.

Minors

1. Please define SEC throughout the manuscript.

2. In page 5, line 218, “azide” should be “azides”.

3. In page 5, line 219, add “of” after “instead”.

4. In page 7, caption of figure 2, add “monomer” before “conversion”.

--> According to your suggestions, we have revised the sentences as follows;

Page 3, lines 108-109;

“Analytical size exclusion chromatography (SEC)”

Page 6, line 226;

Size exclusion chromatography (SEC) analysis”

Page 5, lines 218-219;

“with the corresponding disaccharide azides.”

Page 5, line 220;

“the VE versions instead of using N-propargyl MI.”

Page 7, caption of figure 2;

“Figure 2. Time-monomer conversion curves”

Round 2

Reviewer 2 Report

The manuscript was improved according to the suggestion. I think this work can be accepted to publish in Polymer.